# A comparison of two remotely operated vehicle (ROV) survey methods used to estimate fish assemblages and densities around a California oil platform

Milton S. Love[1]☯*, Mary M. Nishimoto[1]☯, Scott Clark[1]☯, Li Kui[1]☯, Azivy Aziz[2]‡, David Palandro[2]‡

1 Marine Science Institute, University of California, Santa Barbara, California, United States of America,
2 ExxonMobil Upstream Research Company, Spring, Texas, United States, United States of America

☯ These authors contributed equally to this work.
‡ These authors also contributed equally to this work.
* love@lifesci.ucsb.edu

**Data Availability Statement:** Love, M., M. Nishimoto, and L. Kui. 2020. Data to support manuscript: A Comparison of Two ROV Survey

## Abstract

Offshore oil and gas platforms have a finite life of production operations. Once production ceases, decommissioning options for the platform are assessed. The role that a platform's jacket plays as fish habitat can inform the decommissioning decision. In this study, conducted along the crossbeams of a California platform jacket and using an ROV, we compared estimates of fish diversity and densities determined from a targeted "biological" survey with those from a replicated "structural" survey. We found that the *water column* fish species assemblages characterized by the two methods were similar. By contrast, the two survey methods yielded different species assemblages inhabiting the crossbeam at the platform jacket *base*. This difference occurred because, at least off California, the platform jacket base species diversity tends to be highest where the bottom crossbeam is undercut, creating sheltering sites for many species. Because the structural method inadequately imaged the seafloor-crossbeam interface, particularly where a gap occurred between crossbeam and seafloor, substantial numbers of fishes were not visible. While we cannot extrapolate from this study to all platforms' worldwide, it is clear that routine platform structural integrity surveys may be a valuable source for opportunistic marine community surveys. Intentional planning of the structural survey to incorporate relatively minor variations (e.g., maintaining fixed ROV distance from the infrastructure and consistent 90˚ camera angle) coupled with a deliberate consideration of the platform ecology (e.g., positioning the ROV to capture the seafloor-crossbeam interface) can substantially improve the effects on fish assemblage assessments from routine structural surveys without compromising the integrity assessment. We suggest that these biases should be both acknowledged and, understood when using routine structural surveys to inform platform ecology assessment. Additional consideration may be given to structural surveys that incorporate incremental adjustments to provide better data applicability to biological assessments.

Methods Used to Estimate Fish Assemblages and Densities Around a California Oil Platform ver 1. Environmental Data Initiative. https://doi.org/10.6073/pasta/5f2d77235388717672ff612cc7fa7d7c.

**Funding:** This study was funded by an ExxonMobil Upstream Research Company contract to the University of California, Santa Barbara (UCSB), contract number EM11487 to M.L. and M.N. Authors Azivy Aziz and David Palandro are employed by ExxonMobil Upstream Research Company. ExxonMobil Upstream Research Company provided support in the form of salaries for authors AA and DP, but did not have any additional role in the study design, data collection and analysis, decision to publish, or preparation of the manuscript. The specific roles of these authors are articulated in the 'author contributions' section.

**Competing interests:** This research was funded by an ExxonMobil Upstream Research Companycontract to the University of California, Santa Barbara (UCSB),contract number EM11487. A. Aziz and D. Palandro are employees of ExxonMobil Upstream Research Company. M. Love, M. M. Nishimoto, S. Clark, and L. Kui were employed by UCSB under that contract. There are no patents, products in development or marketed products to declare. This does not alter our adherence to all the PLOS ONE policies on sharing data and materials

## Introduction

Each of the thousands of offshore oil and gas platforms worldwide has a finite production life. Once a decision is made to cease production, governmental agencies undertake a decommissioning process to decide on the disposition of that platform jacket (most often either partial or total removal, [1]). The role that a platform's jacket plays as fish and invertebrate habitat can be part of that decision-making process [1], and the increasing number of platforms to be decommissioned increases the need for this information. Historically, these characterizations have come from targeted biological surveys (e.g., conducted by scuba divers, manned submersibles, remotely operated vehicles (ROVs), bioacoustics, and nets) of a limited number of offshore oil and gas structures [2–7]. However, there also exists oil and gas ROV engineering surveys of platforms and associated infrastructure (i.e., pipelines, subsea equipment and wellheads) conducted as part of routine physical integrity inspections. In the past few years, researchers off western Australia [8–12] and in the North Sea [13, 14] have begun to use this archival footage to characterize the biological communities associated with the offshore oil and gas structures.

This relatively recent development has led to discussions on the value of using industry structural surveys for biological assessments [12, 15, 16]. While this research makes clear that structural surveys can be useful in surveying marine life around platform jackets, there remain questions regarding the potential biases of this methodology [15, 16]. Thus, there is a need for studies that compare biological data from targeted biological surveys with those taken to assess structural integrity. In this study, using an ROV, we conducted a pilot project that compared estimates of fish diversity and densities determined from a targeted biological survey with those from a replicated structural survey at a platform jacket.

## Materials and methods

Surveys were conducted at the ExxonMobil Platform Harmony located in the Santa Barbara Channel (34˚22'N, 120˚10'W), southern California (Fig 1). Harmony was installed in 1989, is 10.3 km from shore, and sits at a bottom depth of about 363 m [17]. To reduce possible variation, the two comparison surveys were conducted by the same ROV, at a single platform, over the same depths, and on the same days. We used a work-class *Comanche*-type ROV for these surveys. All surveys were conducted at the same speed (about 0.5 knots) using a SubC 1CamMk5 HDf video camera and lights at all depths. The research was conducted along crossbeams during daylight hours on 25–26 August 2018 at water column depths of 17 m, 38 m, 61 m, and 182 m, and at the bottom-most crossbeam at 363 m (crossbeam lengths are shown in Table 1). At each depth, we surveyed the north, west, and south sides of the platform jacket (i.e., the east side was not surveyed). All surveys were conducted during daylight hours, and decisions regarding which side was surveyed first, which method was first used on a specific side, and the length of time between surveys on a specific side were made haphazardly (Table 2).

Sequence of crossbeam surveys, "structural" and "biological," conducted at Platform Harmony. Included are 1) crossbeam depth, 2) crossbeam designation, 3) order of the two surveys (structural or biological) at a given crossbeam, 4) time of survey, and 5) time between surveys at a specific crossbeam.

Using two survey methods, that we termed "biological" and "structural," we compared the densities and diversities of fishes associated with these crossbeams. The *biological* surveys were based on methodologies we have utilized when surveying fishes around California platforms using both manned submersibles [2] and an ROV [18, 19]. In the biological method, the ROV traveled parallel to a platform crossbeam, the camera was aimed at a 90˚ angle to that

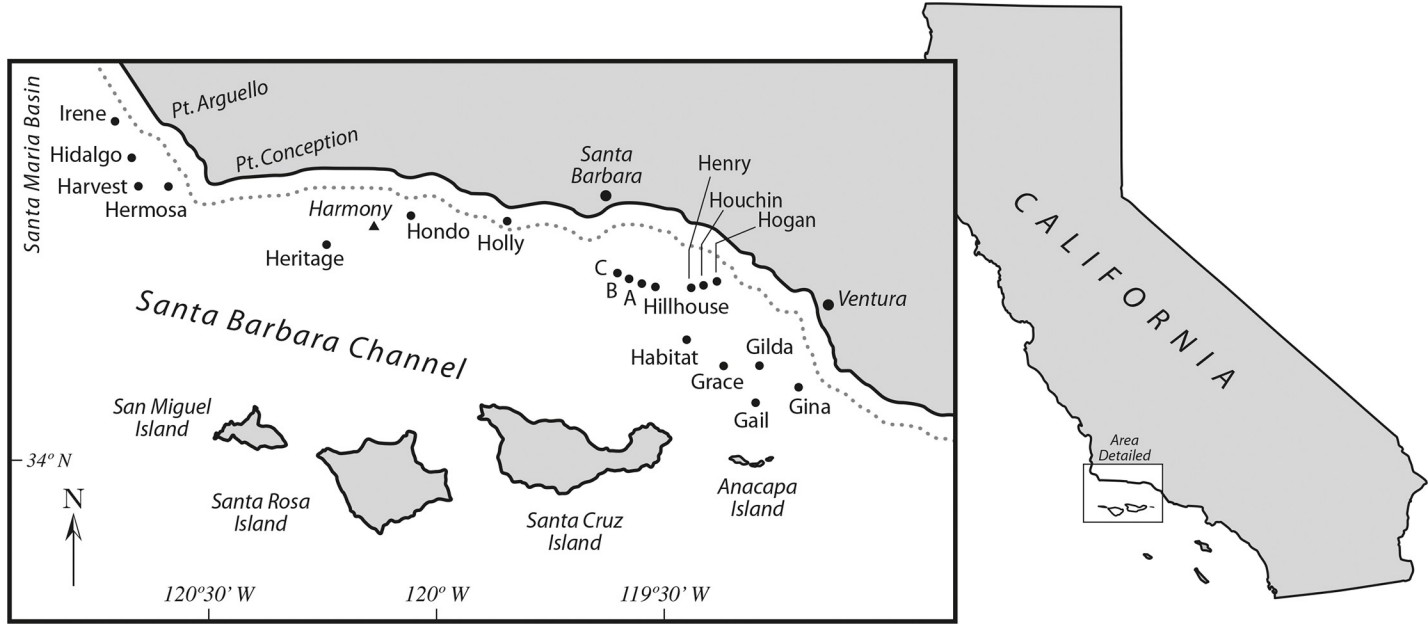

**Fig 1. Location of Platform Harmony, Santa Barbara Channel, southern California.**

crossbeam, and the ROV remained about 2 m from the structure (Fig 2). Importantly, when surveying the bottom-most crossbeam (which sat on or just off the seafloor), both the seafloor bottom-crossbeam interface and any crossbeam undercut were kept in the field of view. This was because research has shown that off California, fishes dwelling around platform bases tend to associate with those portions of the bottom crossbeam that are undercut (creating a crevice) rather than those sections where there is no gap [19]. We note that in the biological method, the upper part of the bottom crossbeam was sometimes not visible.

The *structural* technique was designed to replicate an industry engineering platform inspection survey. The goal of structural inspections is to examine the oil and gas infrastructure including the jacket crossbeams, for structural integrity issues. When using the structural method, we requested that the ROV pilot, who had conducted many ROV structural platform surveys, conduct a survey of the crossbeams as if he were conducting a typical structural survey. The major differences between the two methods were 1) during the structural survey the pilot tended to angle the ROV such that, as the ROV moved forward, the pilot could see what was ahead as the ROV traversed a crossbeam (Fig 2). In practice, this meant that the camera

**Table 1.**

| Crossbeams included in 2018 ROV surveys | | |
|---|---|---|
| Depth (m) | North and South (m) | West (m) |
| 17 | 46.0 | 28.9 |
| 38 | 50.3 | 32.6 |
| 61 | 54.9 | 36.6 |
| 182 | 79.2 | 58.1 |
| 363 | 117.3 | 91.6 |

Lengths of Platform Harmony crossbeams surveyed in this study. The east crossbeam was not surveyed.

**Table 2.**

| Depth (m) | Side of platform | 1st pass | 2nd pass | 1st pass ends | 2nd pass begins | Elapsed time between passes | Notes |
|---|---|---|---|---|---|---|---|
| 17 | North | 1 | 4 | 15:35:53 | 16:08:44 | 0:32:51 | "Pass" refers to transect number |
| 17 | West | 2 | 5 | 15:42:03 | 16:30:00 | 0:47:57 | Color coding: |
| 17 | South | 3 | 6 | 15:50:00 | 16:44:29 | 0:54:29 | Structural survey |
| 38 | North | 30 | 35 | 17:12:02 | 17:55:35 | 0:43:33 | Biological survey |
| 38 | West | 31 | 34 | 17:20:05 | 17:49:05 | 0:29:00 | |
| 38 | South | 32 | 33 | 17:29:58 | 17:35:26 | 0:05:28 | |
| 61 | North | 24 | 25 | 15:27:02 | 15:30:25 | 0:03:23 | |
| 61 | West | 26 | 27 | 15:50:50 | 15:53:46 | 0:02:56 | |
| 61 | South | 28 | 29 | 16:17:25 | 16:19:45 | 0:02:20 | |
| 183 | North | 18 | 19 | 12:51:30 | 12:55:45 | 0:04:15 | |
| 183 | West | 20 | 21 | 13:24:16 | 13:27:36 | 0:03:20 | |
| 183 | South | 22 | 23 | 14:03:59 | 14:06:01 | 0:02:02 | |
| 372 | West | 8 | 12 | 19:01:45 | 22:07:00 | 3:05:15 | |
| 372 | North | 9 | 11 | 19:26:57 | 20:25:30 | 0:58:33 | |
| 372 | South | 10 | 13 | 20:01:05 | 22:32:37 | 2:31:32 | |

angle to the crossbeam varied but was centered around 45˚ (compared to a camera angle of 90˚ during the biological surveys). 2) The distance from the ROV to the jacket was variable and was often closer to the crossbeam (primarily 1 m or less) than in the biological inspection. 3) Importantly when surveying the *bottom-most* crossbeam (immediately adjacent to the seafloor), the pilot tended to remain somewhat above the seafloor, higher than for the biological surveys. Thus, compared to the biological method, in the structural survey the bottommost crossbeam was viewed from a higher vantage and the seafloor-crossbeam interface was sometimes not visible and the crevice under that crossbeam (if present) was never visible. All footage, along all crossbeams, was included in the analyses.

## Statistical analysis

We identified fish taxa to the lowest taxonomic level possible. Due to the challenges on distinguishing some fish species at their young-of-year stage, these unidentified young-of-year fish were grouped as one species in the following analysis. Therefore, herein, our use of the term "species" refers not only to single species but also to species aggregates. Species density was calculated as the number of individuals for a given species on a given transect divided by the length of the transect. To better visualize the density values in figures, we multiplied the density by 100 to obtain density in the unit of number per 100 $m^2$.

We examined the effect of the survey methods (biological and structural methods) on fish density at each of the habitats (midwater and base). Mixed-model ANOVA was used to statistically test whether the difference in fish density was driven by the survey methods, and the depth of crossbeam was a random factor to account for fish density variance in the water column.

To visualize the relationships between species assemblages over habitats and survey methods, we created a two-dimensional, non-metric multidimensional scaling (nMDS) plot using the "metaMDS" function in the "vegan" package in R [20]. The sample matrix in the NMDS analysis was was comprised of the densities of the top ten most abundant species that consisted of 83% of the total fish count.

The resulting stress value of 0.08 indicated that the reduced dimensions well represented the original community assemblage. To statistically test whether the species assemblages

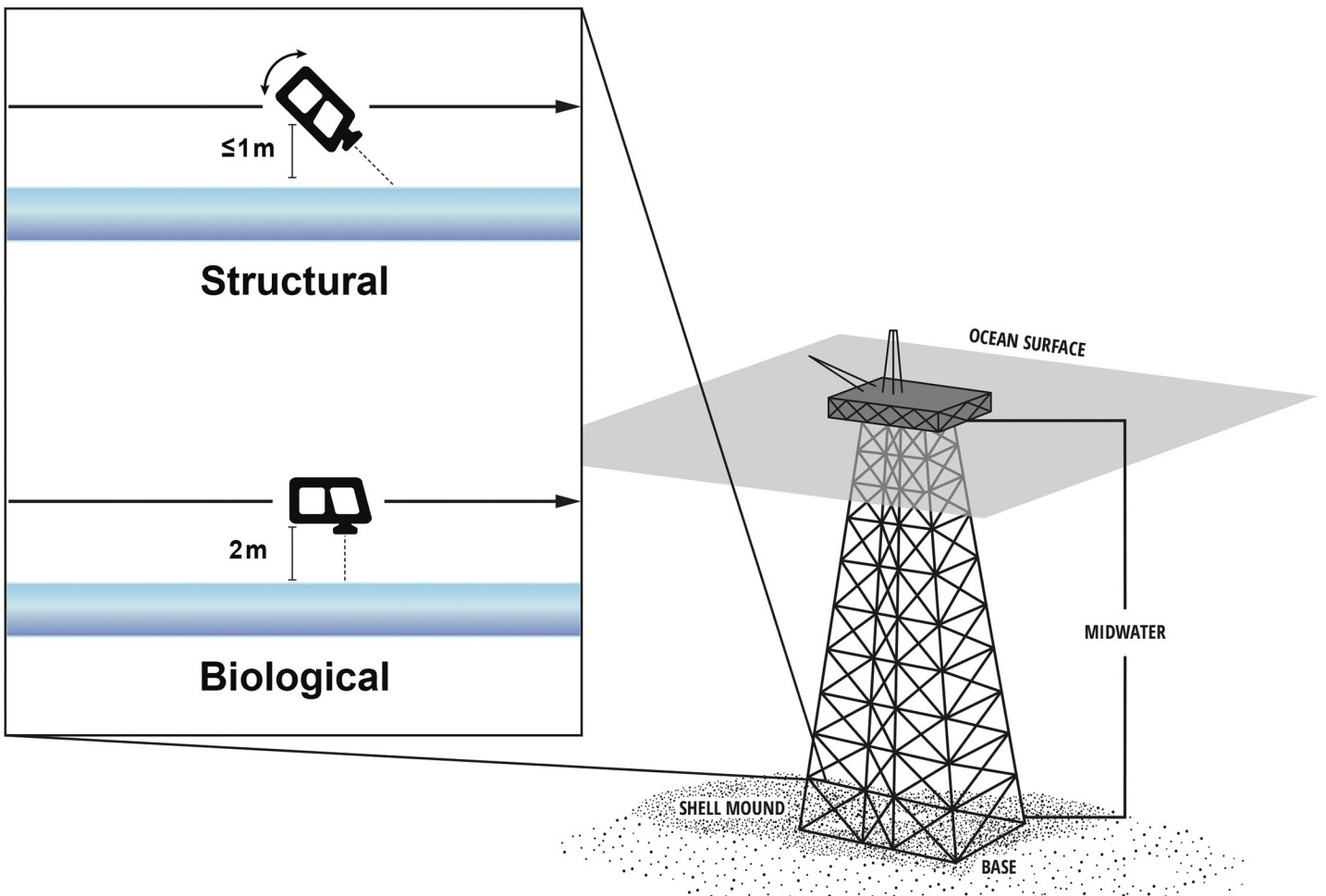

**Fig 2. A downward looking view of the orientation of the ROV along horizontal crossbeams in "biological" and "structural" surveys.** In the biological surveys, the camera was oriented at a 90˚ angle to the crossbeam and the ROV was kept at about 2 m from the jacket. In a simulation of structural surveys, the camera angle to the crossbeam varied but was centered around 45˚ and the ROV was usually less than 1 m from the crossbeam.

between survey methods and habitats were different, we conducted the Analysis of Similarities (ANOSIM), the anosim() function in "vegan" package. Both nMDS and ANOSIM were performed on the Bray-Curtis dissimilarity indices from the sample matrix using vegdist() function.

## Results and discussion

### Platform jacket midwaters

The midwater species assemblages characterized by the two methods were similar (Fig 3). We observed a minimum of 19 fish species using the biological method and 18 species with the structural technique amid crossbeams between 17 m and 182 m (Table 3). Two "species", *Sebastes flavidus/serranoides* and *Rhinogobiops nicholsii* (along with a handful of unidentified fishes), were only observed using the biological method, and two species, *Sebastes carnatus* and *Sebastes serriceps* were unique to the structural surveys. All were observed in very small numbers. Among the commercially important taxa observed were the squarespot rockfish, *Sebastes hopkinsi*, unidentified young-of-the-year (YOY) rockfishes, painted greenling,

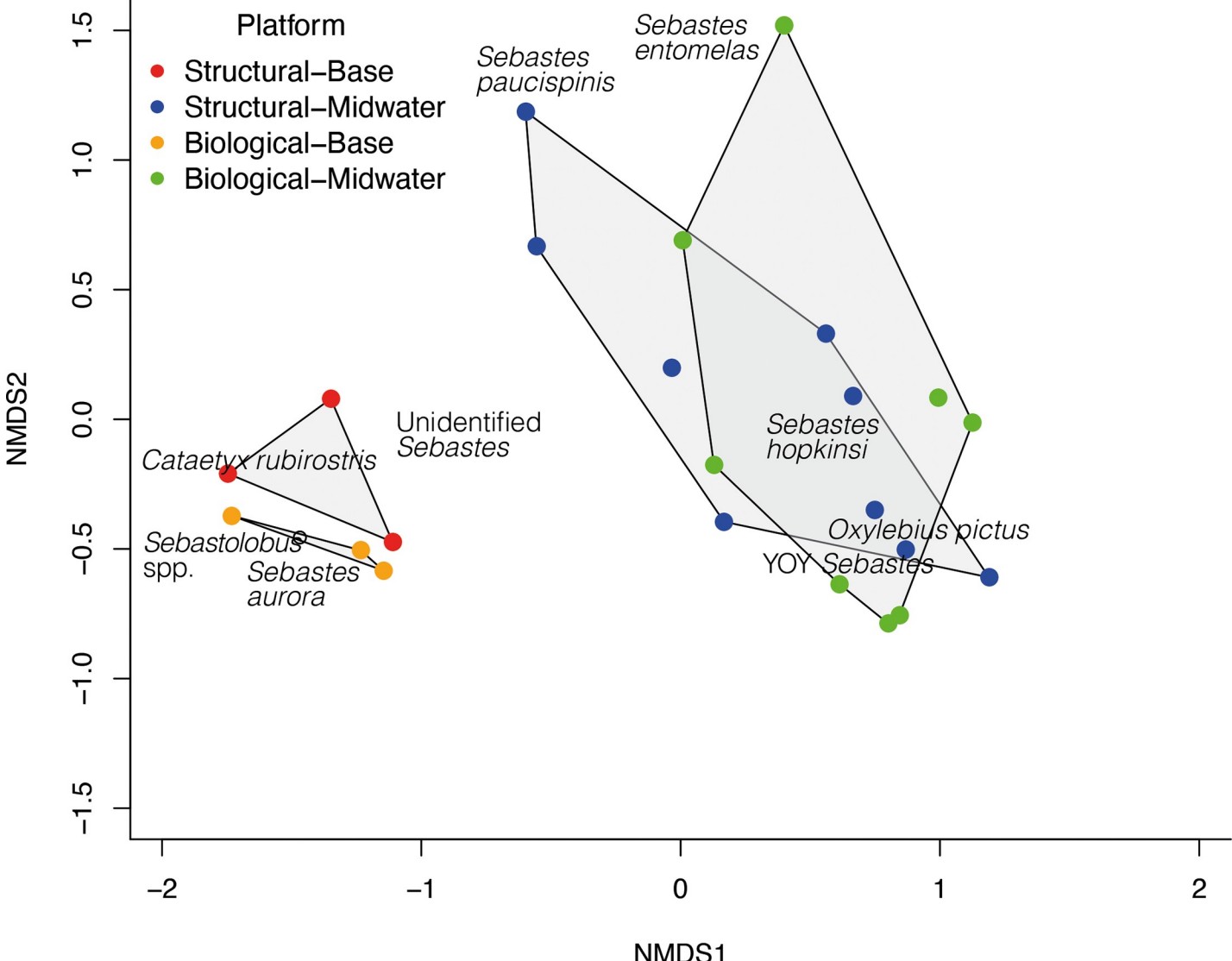

**Fig 3. NMDS plot of the 10 most abundant species (83% of all fishes) observed at Platform Harmony from the ROV surveys.**

*Oxylebius pictus*, unidentified rockfishes, *Sebastes* spp., blue rockfish, *Sebastes mystinus*, and widow rockfish, *Sebastes entomelas*. Regardless of the protocol, the majority of taxa observed were rockfishes (15 of 19 species for the biological method and 14 of 18 species for the structural method), and rockfishes dominated by both numbers (266/299 and 181/223, respectively) and densities (90.8% and 79%, respectively) (Table 2).

Number and densities (number per 100 m$^2$) of fishes observed in the midwaters of Platform Harmony by two ROV survey methods, "biological" and "structural" (these are defined under Methods). YOY–young-of-the-year.

Although mean densities derived from the biological method were higher than from the structural method (35.2 m$^{-2}$ and 26.9 m$^{-2}$, respectively), they were not statistically different (p >0.05) (Table 3, Fig 4). Particularly high densities occurred at three crossbeams. When we more closely examined the video footage, we found that these extremely high densities were due to loose and moving aggregations of young rockfishes. These aggregations were present

**Table 3.**

| Species | Method | | | | |
|---|---|---|---|---|---|
| | Biological | | Structural | | |
| | Number | Density | Number | Density | |
| *Sebastes hopkinsi* | 93 | 11.5 | 29 | 3.9 | |
| YOY *Sebastes* | 87 | 10.9 | 44 | 5.5 | |
| *Oxylebius pictus* | 22 | 2.8 | 36 | 4.6 | |
| *Sebastes* spp. | 20 | 2.0 | 8 | 0.7 | |
| *Sebastes mystinus* | 14 | 1.4 | 3 | 0.3 | |
| *Sebastes entomelas* | 14 | 1.1 | 47 | 6.2 | |
| *Sebastes dallii* | 8 | 0.8 | 5 | 0.5 | |
| *Sebastes caurinus* | 6 | 0.7 | 11 | 1.3 | |
| *Sebastes paucispinis* | 5 | 0.4 | 13 | 0.9 | |
| *Sebastes rubrivinctus* | 5 | 0.5 | 4 | 0.4 | |
| Unidentified fishes | 5 | 0.6 | 0 | 0.0 | |
| *Sebastes saxicola* | 4 | 0.5 | 5 | 0.6 | |
| *Sebastes miniatus* | 4 | 0.4 | 1 | 0.2 | |
| *Chromis punctipinnis* | 2 | 0.3 | 2 | 0.2 | |
| *Scorpaenichthys marmoratus* | 2 | 0.2 | 2 | 0.3 | |
| *Sebastes semicinctus* | 2 | 0.3 | 1 | 0.2 | |
| *Sebastes flavidus/serranoides* | 2 | 0.3 | 0 | 0.0 | |
| *Rhinogobiops nicholsii* | 1 | 0.2 | 0 | 0.0 | |
| *Sebastes atrovirens* | 1 | 0.1 | 6 | 0.7 | |
| "KGB" YOY[1] | 1 | 0.1 | 2 | 0.2 | |
| *Sebastomus* sp. | 1 | 0.1 | 2 | 0.2 | |
| *Sebastes carnatus* | 0 | 0.0 | 1 | 0.1 | |
| *Sebastes serriceps* | 0 | 0.0 | 1 | 0.1 | |
| Total | 299 | 35.2 | 223 | 26.9 | |
| Total number of species | 19 | | 18 | | |
| Percent *Sebastes* by number | 90.0 | | 81.2 | | |

[1]*Sebastes atrovirens*, *Sebastes carnatus*, *Sebastes caurinus*, and/or *Sebastes chrysomelas*.

during one of two passes along the crossbeams (Points A, B, and C in Fig 4), each pass being a different survey method, and densities along the same crossbeams were substantially lower on the other. Importantly, Point A represents a second pass along a crossbeam using the structural method following a first pass using the biological method when lower density was observed, and Points B and C, first passes using the biological method when densities were higher than the second passes. Therefore, to the extent that we can determine, in these instances differences in densities were not due to the difference in ROV survey method nor to the order by which the two methods were conducted along a crossbeam.

## Platform jacket base

The nMDS plot shows dissimilarity between the two assemblages at the base (Fig 3), although the few samples precluded testing the statistical significance of this difference. Compared to the midwaters, the platform jacket base harbored fewer species (10 for each protocol), although rockfishes were again the dominant group. Important species in the biological survey of the base of the platform jacket included unidentified *Sebastes* (presumably *Sebastes aurora* and/or

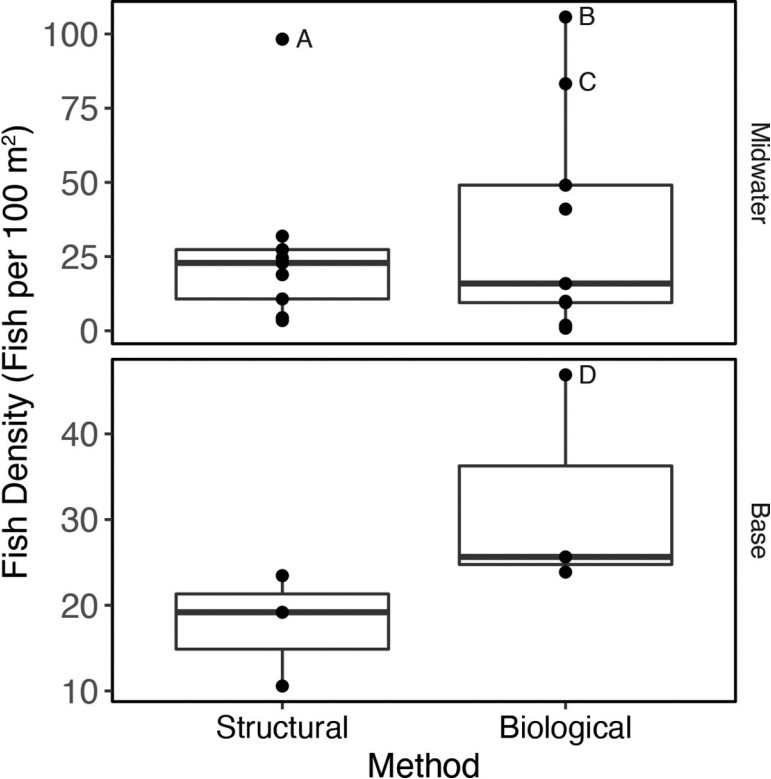

**Fig 4. Median densities and 25% and 75% quantiles of all fishes combined in the midwaters and at the base of Platform Harmony using two protocols, "biological": And "structural".** Each dot represents the density of fishes along a single crossbeam between corners of the platform (n = 12, two sets of two points overlap and appear as one). Fish density values were multiplied by 100 to obtain density units of fish per 100 m². The dots labeled A-D identify outliers.

*Sebastes melanostomus*), *Sebastes melanostomus*, *Sebastes aurora*, and *Sebastes babcocki*, as well as *Sebastolobus* spp. (Fig 4, Table 4). The important species in the structural survey at the base were unidentified *Sebastes*, *Cataetyx rubrirostris*, and *S. melanostomus*. Two "species", a single *Eptatretus stouti* and an unidentified Nettastomatidae, were unique to the biological survey and three "species", an unidentified flatfish, a *Sebastomus* sp., and three *Leuroglossus stilbius* individuals, were only see in the structural survey. These were all observed in very small numbers. As in the midwaters, overall mean densities were higher using the biological survey technique than the structural technique (32.1 m$^{-2}$ versus 17.7 m$^{-2}$) although the difference was not statistically significant. A single extreme value accounted for the difference (Point D, Fig 4). This highest fish density was observed on the north side of the platform jacket, on the second pass, using the biological method, and was driven primarily by two species groups: unidentified *Sebastes* with a density of 23 m$^{-2}$ and *Sebastes melanostomus*, with a density of 13 m$^{-2}$. By comparison, the lowest density estimate was observed using the structural method on the north side, and the two survey techniques yielded similar density estimates on the south and west sides of the platform jacket (Fig 5).

Number and densities (number per 100 m²) of fishes observed at the base of Platform Harmony by two ROV survey methods, "biological" and "structural" (these are defined under Methods).

Why would the biological and structural survey methods yield different species assemblages along the crossbeam *base*? And, why would the density of fishes be substantially higher along

**Table 4.**

| Species | Biological | | Structural | |
|---|---|---|---|---|
| | Number | Density | Number | Density |
| Unidentified *Sebastes*[1] | 88 | 14.7 | 58 | 9.7 |
| *Sebastes melanostomus* | 46 | 7.5 | 14 | 2.3 |
| *Sebastes aurora* | 17 | 2.8 | 3 | 0.5 |
| *Sebastolobus* spp.[2] | 17 | 2.5 | 5 | 0.7 |
| *Sebastes babcocki* | 10 | 1.6 | 2 | 0.3 |
| *Cataetyx rubrirostris* | 9 | 1.3 | 15 | 2.3 |
| Unidentified fishes | 4 | 0.6 | 4 | 0.7 |
| *Microstomus pacificus* | 3 | 0.4 | 2 | 0.3 |
| *Parmaturus xaniurus* | 2 | 0.3 | 1 | 0.2 |
| *Eptatretus stouti* | 1 | 0.2 | 0 | 0.0 |
| Unidentified Nettastomatidae | 1 | 0.2 | 0 | 0.0 |
| Unidentified flatfish | 0 | 0.0 | 1 | 0.1 |
| *Leuroglossus stilbius* | 0 | 0.0 | 3 | 0.5 |
| *Sebastomus* sp. | 0 | 0.0 | 1 | 0.2 |
| Total | 198 | 32.1 | 109 | 17.7 |
| Total number of species | 10 | | 10 | |
| Percent Species by number | 81.3 | | 71.6 | |

[1]Likely *Sebastes aurora* and/or *Sebastes melanostomus*.

[2]Likely *Sebastolobus alascanus*.

one side of the platform jacket using the biological survey method? To address these questions, we compared:

A) the structural characteristics of the three crossbeams at the base of the platform jacket (specifically the amount they were undercut);

B) the position of fishes associated with these crossbeams (whether they were associated with the seafloor-crossbeam interface or associated with the sides or tops of the crossbeams);

C) how much of the seafloor-crossbeam interface was visible using the two methods.

The most apparent structural difference among the crossbeams along the three sides of the platform jacket was that the north side was completely undercut as observed in the biological survey which was designed to view the interface between the seafloor and the crossbeam (Fig 6). A continuous crevice was visible beneath the north crossbeam. In comparison, 55% and 13.4% of the crossbeam was undercut on the west side and south side of the platform jacket, respectively (Fig 6).

Both types of survey techniques revealed that fishes were more abundant on the seafloor than above the seafloor along the side and upper surface of crossbeams (Table 5). On all three sides of the platform jacket, a greater proportion of fishes were seen on the seafloor in the biological survey than in the structural survey.

A comparison of the numbers of fishes at each crossbeam (north, west, and south) and the number and percentages of fishes associated either on the "bottom" and near or under the sea floor-crossbeam interface or "off-bottom" along the crossbeam sides and upper surfaces.

Importantly, between the two methods, there was a substantial difference in the amount of the seafloor-crossbeam interface that was visible. The interface along all three crossbeams was

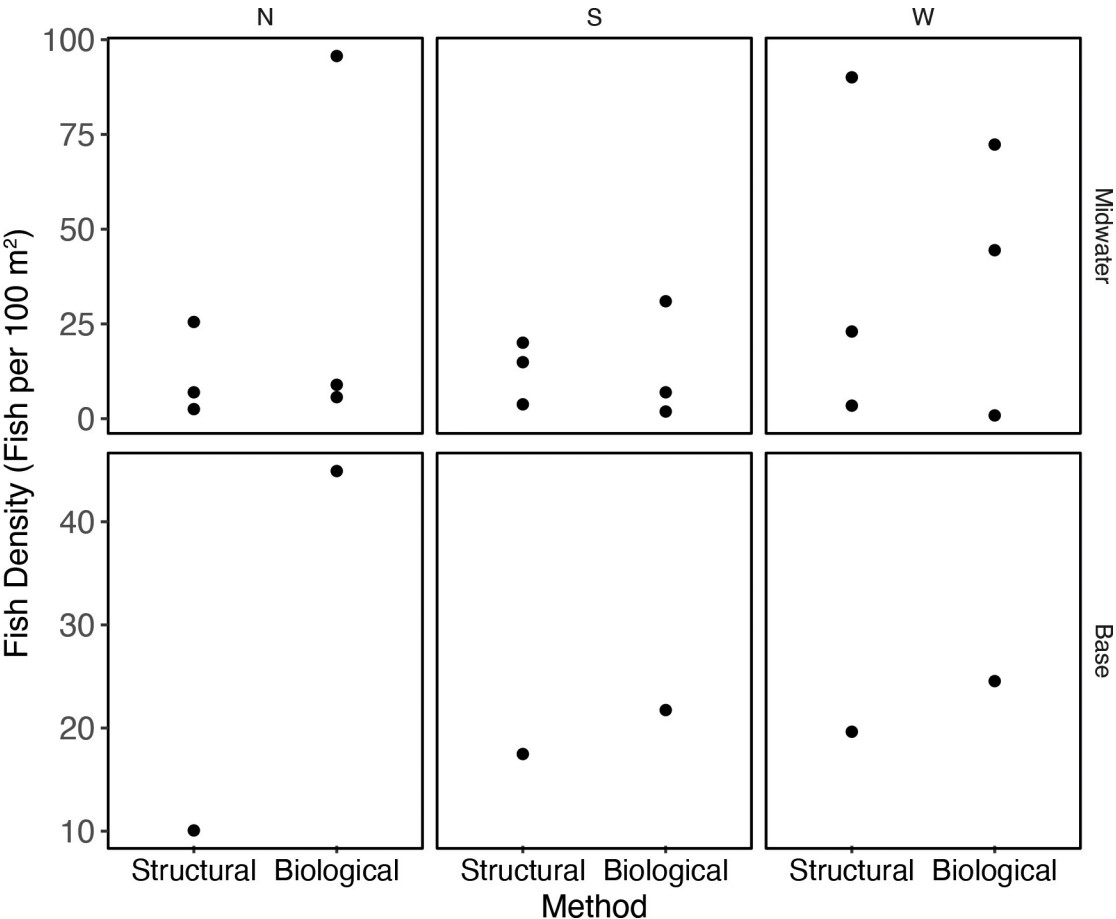

**Fig 5. Densities of all fishes combined in the midwaters and at the base of Platform Harmony, by method and crossbeam side (N = north, S = south, W = west).**

completely visible using the biological method because the ROV was near the seafloor and its camera was aimed directly at the interface. In contrast, the structural survey, where the ROV was positioned higher above the seafloor with the field of view aimed to include the top and side of the crossbeam, the amount of the seafloor-crossbeam interface visible varied between 8.8% of the length of the north beam and 83.0 and 97.0% of the lengths of the west and south beams, respectively (Fig 6). This difference between the north beam and the other two crossbeams occurred because in the structural method the seafloor was visible only where the crossbeam was sufficiently buried such that both the top of the crossbeam and the seafloor-crossbeam interface were within the field of view. In contrast, from this vantage of the ROV, the undercut area if present beneath the crossbeam was obscured from view by the beam itself, and consequently the fishes in the undercut area below the crossbeam would not be visible (Fig 7).

Previous surveys at other California platform jackets have demonstrated that some species occur at greater densities in areas undercut below a crossbeam than adjacent areas where the cross beam is embedded in the sediment [21]. They termed these species members of the "sheltering habitat guild" and *Sebastes aurora*, one of the species responsible for the differences in assemblages, is a member of that guild. Because the structural method inadequately imaged that seafloor-crossbeam interface, particularly along the north crossbeam, substantial numbers of fishes were obscured from view and not surveyed.

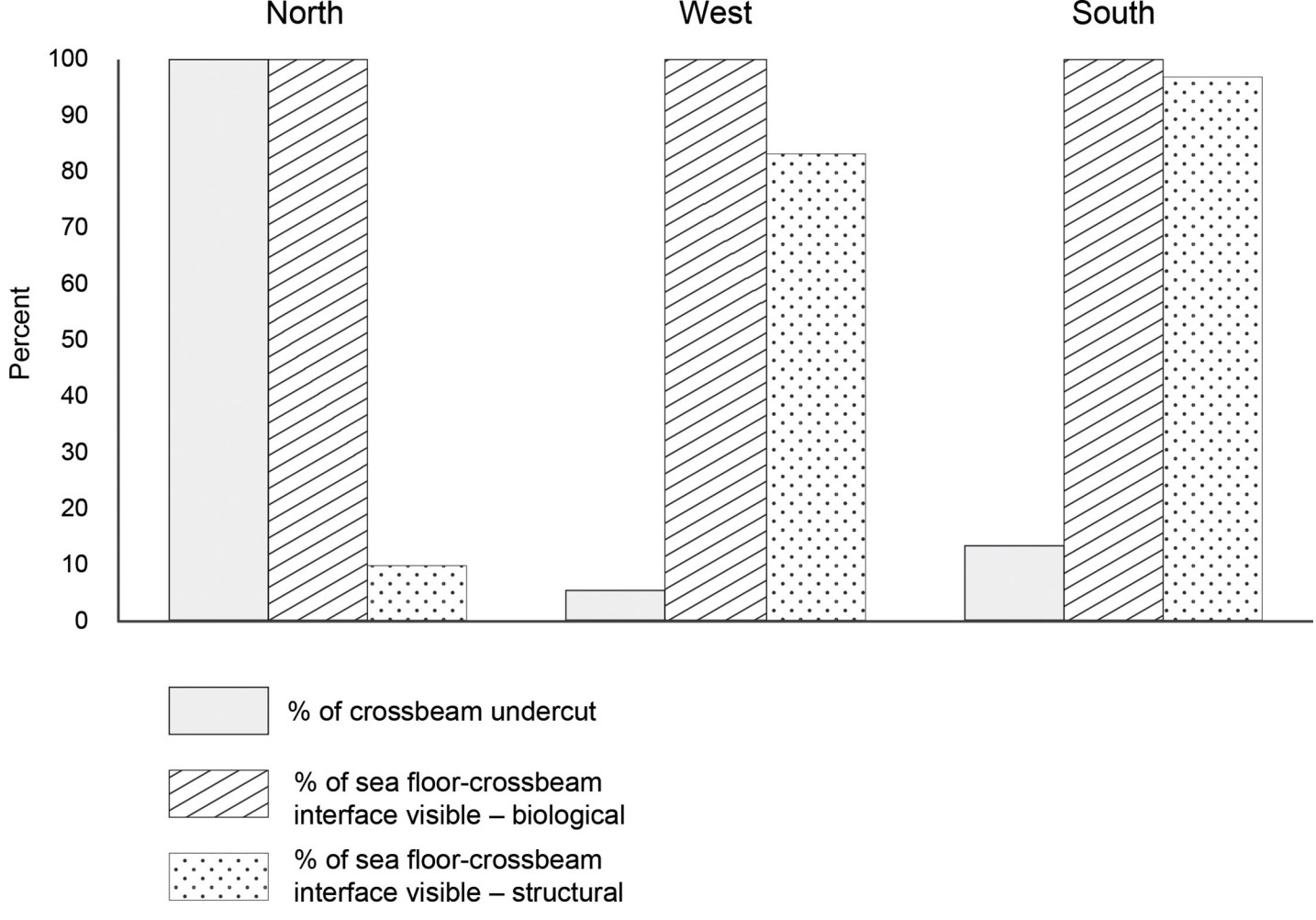

**Fig 6. The percentage of the lengths of the north, west, and south crossbeams that were undercut, and the percentage of the seafloor-crossbeam interface that was visible at each crossbeam using the biological and structural survey method.** The entire length (100%) of the seafloor-crossbeam interface along each crossbeam was visible in the biological survey.

**Table 5.**

| Biological | | |
|---|---|---|
| **North** | **West** | **South** |
| Total number of fishes: 96 | Total number of fishes: 47 | Total number of fishes: 57 |
| Number of fishes on bottom: 84 | Number of fishes on bottom: 36 | Number of fishes on bottom: 49 |
| Number of fishes off bottom: 12 | Number of fishes off bottom: 11 | Number of fishes off bottom: 8 |
| Percent of fishes on bottom: 87.5 | Percent of fishes on bottom: 76.6 | Percent of fishes on bottom: 80.6 |
| **Structural** | | |
| Total number of fishes: 21 | Total number of fishes: 43 | Total number of fishes: 46 |
| Number of fishes on bottom: 12 | Number of fishes on bottom: 24 | Number of fishes on bottom: 31 |
| Number of fishes off bottom: 9 | Number of fishes off bottom: 19 | Number of fishes off bottom: 15 |
| Percent of fishes on bottom: 57.1 | Percent of fishes on bottom: 55.8 | Percent of fishes on bottom: 67.4 |

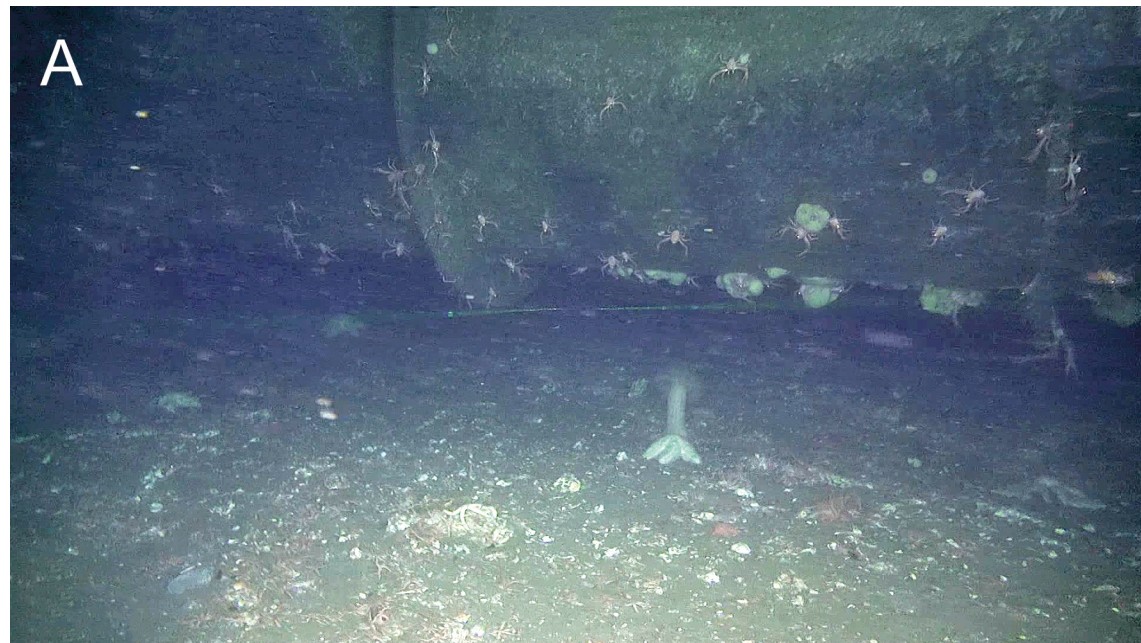

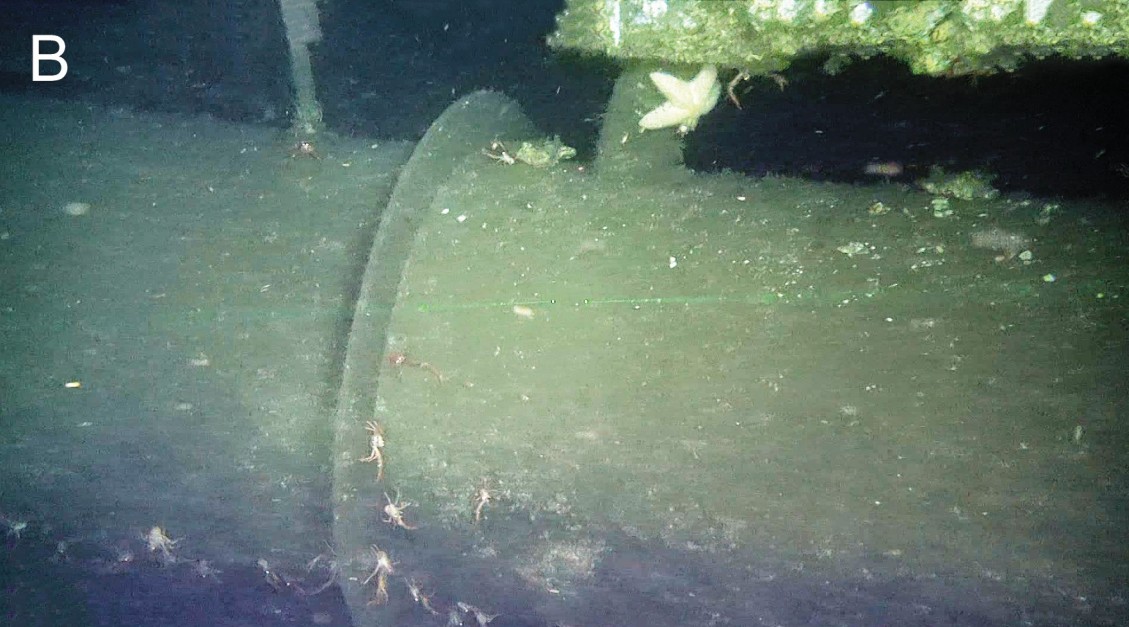

**Fig 7. Two images of the same section of the west crossbeam.** Image (A) was taken during the biological survey and image (B) during the structural survey. Note that in the biological survey, the seafloor-crossbeam interface is visible as is the undercut beneath the crossbeam. In image (B), the seafloor-crossbeam interface is not visible; however, compared to (A) more of the middle and upper part of the crossbeam is visible in (B).

Other studies have raised the potential advantages and limitations of using repurposed structural surveys (e.g., 14–16). As one study noted [16] "Most ROV operations are conducted by industry in a way that fulfills immediate industry requirements but which can confound scientific interpretations of the data. For example, there is variation in video resolution, ROV speed, *distance above the substrate* [italics ours] and time (e.g. both seasonal and time of day)."

Similarly, as other investigators [14] successfully used inspection ROV footage to assess invertebrates and fishes along North Sea pipelines, they too highlighted several aspects that can bias the biological assessments. In particular, and similar to our study, differing camera angles "led to inconsistencies in the section of pipeline assessed for each sample according to each camera view" and "The angle of lateral cameras also influenced the amount of seabed visible either side of the pipeline."

Our study did not address other issues inherent in using ROVs as fish survey tools [10, 16, 22]. As an example, a survey off central California [22] compared the reactions of fishes between a human-operated vehicle (HOV) and an ROV on natural habitats. It found that significantly more fishes reacted (by swimming away) from an ROV than an HOV. The authors speculated that these differences could be due to the presence of a tether on the ROV (absent from the HOV), differences between the ROV and HOV in the positioning of lighting (forward versus starboard, respectively), difference in sounds emitted by the vehicles, and the difference in the size of the vehicles. Following up on this work [23], these researchers compared fish diversities, densities, and sizes obtained during this same study. They found that 1) density estimates of taxa associated with the seafloor were higher in the HOV surveys, 2) a greater percentage of HOV-observed fishes could be sized, and 3) a higher percentage of fishes in the HOV surveys could be identified to species.

Due to the small sample size (n = 12) for each of the survey methods and the short survey time (2 days), only simple statistics were computed in this study to explore the overall difference between these two survey methods. We note that we cannot extrapolate from this study limited to one platform jacket to surveys of all of California platform jackets, or to platform jackets worldwide, as platform jacket structure, fish behaviors, and ROV operator behaviors are likely to be quite variable. In addition to variances introduced by distance to infrastructure and camera angle, biological assessments from structural surveys can be biased if organisms tend to associate with specific, localized features of a platform jacket and if these features are poorly sampled by the structural surveys. In this study, in the *midwaters* of Harmony, there was no evidence that fishes disproportionately associated with particular parts of the crossbeams and, as importantly, during both biological and structural surveys the span of the diameter of the crossbeam remained in view. Thus, species assemblages and fish densities were similar using the two methods. In contrast, it is clear that, at the base of the platform, fishes were more likely to 1) associate with the seafloor-crossbeam interface and, more particularly, they tended to associate with those portions of the crossbeam that were undercut. During the structural survey, the ROV was not routinely positioned to image the seafloor-crossbeam interface, and thus a substantial number of the fishes were obscured from view that were observed in the biological survey.

This study suggests that structural inspection surveys can be a valuable tool in assessing fish assemblages associated with platform jackets. Integrating a more robust biological methodology for structural surveys may be considered to provide guidance to ROV operators. For instance, a recent study [24] highlighted changes to ROV survey methodology, related to lighting and camera operations, that provided a more complete picture of the biological community around a platform in the Gulf of Mexico. To limit costs, incremental adjustments to regular structural surveys should be considered. Guidance should include the use of HD cameras, maintaining a fixed distance from the platform, speed of the ROV (when ROV is in transit between different structural survey points of interest) and camera angle. In our study, a camera angle positioning the ROV at the seafloor-crossbeam interface to capture the fish assemblages would have proved valuable. These minor adjustments would be expected to increase the data applicability of the already valuable structural surveys to be more in line with targeted biological surveys.

## Supporting information

**S1 File.**
(DOCX)

## Acknowledgments

We thank Monica Pessino for preparing the figures and tables and Jennifer Klaib and Jeff
Kowalishen for their assistance in conducting the field research.

## Author Contributions

**Conceptualization:** Milton S. Love, Mary M. Nishimoto.

**Data curation:** Mary M. Nishimoto, Scott Clark, Li Kui.

**Formal analysis:** Milton S. Love, Li Kui.

**Funding acquisition:** Milton S. Love, Mary M. Nishimoto, Azivy Aziz, David Palandro.

**Investigation:** Scott Clark.

**Methodology:** Milton S. Love, Mary M. Nishimoto, Li Kui.

**Project administration:** Milton S. Love.

**Writing – original draft:** Milton S. Love, Mary M. Nishimoto.

**Writing – review & editing:** Milton S. Love, Mary M. Nishimoto, Scott Clark.

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
