## [Decision Letter · Decision Letter 0]

29 Jun 2020

PONE-D-20-16316

A Comparison of Two ROV Survey Methods Used to Estimate Fish Assemblages and Densities Around a California Oil Platform

PLOS ONE

Dear Dr. Love,

Thank you for submitting your manuscript to PLOS ONE. After careful consideration, we feel that it has merit but does not fully meet PLOS ONE’s publication criteria as it currently stands. Therefore, we invite you to submit a revised version of the manuscript that addresses the points raised during the review process.

You have received 2 reviews, which have raised some problems with this paper, but are generally positive.  I am therefore inviting a major revision, in which all then points raised by the reviewers must be addressed in the revised paper, or in a letter of rebuttal.

We look forward to receiving your revised manuscript.

Kind regards,

Maura (Gee) Geraldine Chapman, PhD DSc

Academic Editor

PLOS ONE

Additional Editor Comments:

Academic Editor

You have received 2 reviews, which have raised some problems with this paper, but are generally positive. I am therefore inviting a major revision, in which all then points raised by the reviewers must be addressed in the revised paper, or in a letter of rebuttal.

3. Thank you for providing the following Funding Statement: 

'Funded by an ExxonMobil Upstream Research Company contract to the University of California, Santa Barbara (UCSB), contract number EM11487 to M.L. and M.N.

The funders had no role in study design, data collection and analysis, decision to publish, or preparation of the manuscript.'

We note that one or more of the authors is affiliated with the funding organization, indicating the funder may have had some role in the design, data collection, analysis or preparation of your manuscript for publication; in other words, the funder played an indirect role through the participation of the co-authors.

a. If the funding organization did not play a role in the study design, data collection and analysis, decision to publish, or preparation of the manuscript and only provided financial support in the form of authors' salaries and/or research materials, please review your statements relating to the author contributions, and ensure you have specifically and accurately indicated the role(s) that these authors had in your study in the Author Contributions section of the online submission form. Please make any necessary amendments directly within this section of the online submission form.  Please also update your Funding Statement to include the following statement: “The funder provided support in the form of salaries for authors [insert relevant initials], but did not have any additional role in the study design, data collection and analysis, decision to publish, or preparation of the manuscript. The specific roles of these authors are articulated in the ‘author contributions’ section.”

If the funding organization did have an additional role, please state and explain that role within your Funding Statement.

Reviewers' comments:

Reviewer's Responses to Questions

**Comments to the Author**

1. Is the manuscript technically sound, and do the data support the conclusions?

Reviewer #1: Yes

Reviewer #2: Partly

2. Has the statistical analysis been performed appropriately and rigorously? 

Reviewer #1: No

Reviewer #2: No

3. Have the authors made all data underlying the findings in their manuscript fully available?

Reviewer #1: Yes

Reviewer #2: Yes

4. Is the manuscript presented in an intelligible fashion and written in standard English?

Reviewer #1: Yes

Reviewer #2: Yes

5. Review Comments to the Author

Reviewer #1: Review of PONE-D-20-16316 A comparison of two ROV survey methods used to estimate fish assemblages and densities around a California Oil Platform

General:

The paper provides a succinct assessment of how dedicated biological surveys of fish communities on a platform jacket compare to data that could be derived from typical industry surveys. The results are valuable as they demonstrate the utility of industry surveys and how slight adjustments to protocol can yield important biological data. It is also rare for such surveys to be conducted, with an effective doubling of sampling effort required by industry to undertake this research. I note the paper is focussed on fish, but examination of epibiota may also yield interesting results.

The paper is generally well written and clear with just a few grammatical errors (see below). I have made some specific suggestions below to improve the manuscript. In particular, further explanation of the differences in methodologies is required and the resulting implications for the study.

Specific:

The addition of a Figure that shows the reader what the jacket structure looks like (surface-seabed with crossbeams) would be a good addition and could be incorporated into Figure 2.

Table 1: Include crossbeam length into the text within the table to make this clear.

Line 88 – Add a sentence explaining why these depths were chosen. Is this the only depths that crossbeams were evident? There is a big gap between 61 and 182 and 182 and the seabed.

Methods in general: I would like to see more information on the differences between the methods. Did the structural survey use lights? How did the speed compare between both surveys? Also, would a structural survey undertake transects in the same manner as that used by the biological survey, i.e. of these same components or did they only survey these same crossbeams for comparison to the biological surveys? This is important because structural surveys may not survey the entire jacket but just sections of interest to industry.

Line 104 - A 2 m distance for a biological survey (focussed on fish) seems very close. McLean et al. (2019) used 5 m to provide an optimal view of the fish assemblage around the jacket. Do you feel that you missed larger species?

Line 138: I would stick with the raw numbers here rather than inflating by x100. Just adjust the scales on Figures so that the pattern is evident and present as per m2. This would mean altering Figures 4, 5 to present the correct values per m2. I really cant see this inflation as being necessary.

Line 145 – missing a word “..we created a two-dimensional..”

Line 147: spelling. Abundance should be abundant.

Table 2: Add “structural” to the industry column for consistency with text. Same for Table 3 and Table 4. I also note that the figures present structural first then biological. You could keep the tables in the same order.

Results – add that three species were uniquely observed by the biological survey and two by the structural survey, albeit each in very low numbers.

Line 180 – should this be 26.9 not 27.1? (to match Table 2).

Line 183: So here you are stating that whichever survey method occurred first was more likely to view these aggregations of young rockfishes. Likely because they were scared away or avoided the ROV subsequently. The following sentence needs further justification then, i.e. for each of the three sides of the jacket or days did you start with a different survey method? (randomised order).

Figure 4 legend: Make it clear in the legend that densities have been inflated (x100)

Line 202 – make sure numbers match the table. They are slightly off.

Line 204 – do you mean that the highest density estimate was obtained on the north side using the biological method? This is unclear here? If referring to Point D, what is the actual species that is driving this? Is it Sebastes melanostomus?

Line 205 – change industrial to “structural”

Table 4 legend is missing a close bracket.

Line 284: missing word. “It showed that..”

Line 293: Great! Bring on HOV surveys into the future.

Line 295: California’s platform jackets – plurals needed.

Line 310: remove invertebrates from the sentence.

Line 312: This sentence does not make sense or is missing a few words.

Reviewer #2: Love and colleagues compared fish assemblage and density data derived from two types of ROV surveys along an ExxonMobil Platform off the coast of southern California. The goal was to determine if video observations from routine approaches used in the industry could be utilized by scientists to monitor fish populations, as is being done by others elsewhere (e.g., Gulf SERPENT project). The first approach was one used by the industry to examine structural integrity (structure) and involved variable camera angles along cross-beams. The second, used more by scientists (biological), involved a 90 degree scan and incorporated the seabed on the bottom cross-beam. Overall, the authors identified community similarities between the two techniques, suggesting a potential benefit of the industry-based methods, although there were significant differences when considering communities at the base of the platform.

The main weakness of the manuscript is that it is based on so few comparative surveys (n=12) between the two approaches, which are preformed at a single platform, spread across different depths of the water column and different "faces" of the platform jacket. It is understood that these surveys are quite costly to implement, but the lack of statistical rigor should be further emphasized. Further, there is a tremendous amount of detail lacking that could affect the interpret-ability of the results. As the authors themselves admit, fish can become deterred by ROV presence. So, a major question becomes: how were these surveys performed (in terms of timing) and could one have impacted the results of the other (e.g., was biological always before industry)? Was randomization possible? All we are provided with are the dates over which the surveys were completed, and that they were done by the same ROV type. It was also clear that the authors were not well-versed in their community analyses as there are terms used incorrectly, and again were lacking details on how they set up their statistical analysis. I have identified these shortcomings in the attached PDF.

Stylistically, the manuscript is well-written and the figures strongly convey the results in a readable manner. However, there were several typos throughout, which have also been highlighted in my review.

In all, I believe Love et al. have some intriguing preliminary evidence that industry-type ROV surveys could provide meaningful data to be used by scientists for monitoring fish densities and assemblages at oil platforms. I am not sure that the findings are particularly "earth-shattering" given the limited number of surveys in a constrained time-period, and feel this manuscript would probably be better suited as a rapid communication.

Specific Comments (by line numbers):

ABSTRACT

Line 13: Abstract - this is way too long. It shouldn't exceed 300 words

INTRODUCTION

Line 67: Consider adding reference:

Ajemian, M. J., Wetz, J. J., Shipley-Lozano, B., & Stunz, G. W. (2015). Rapid assessment of fish communities on submerged oil and gas platform reefs using remotely operated vehicles. Fisheries Research, 167, 143-155.

Line 79: Strange return

MATERIALS AND METHODS

Line 87: Which survey was first?

Line 112: List camera type and specs

Line 131: Need more details on the chronology of the surveys

Line 131: More details on whether all footage was analyzed or if there was a sub-sampling regime

Line 141: Wouldn't this be just a mixed-model ANOVA?

Line 144-147: Need to state data type used in analysis (density, presence/absence)

Line 147: Replace 'abundance' with "abundant"

Line 150: Change 'Dissimilarity' to "similarity:

Line 151: What type of matrix was used?

RESULTS AND DISCUSSION:

Lines 166-168: The NMDS is simply an ordination; it does not give you a significance level. Is this from the ANOSIM?

Line 169: Change 'MNDS' to "NMDS"

Lines 183-186: This is exactly why the chronology of the surveys needs explanation in the methods.

Line 197: For 'primarily,' did you mean "presumably" - it does not make sense otherwise

Line 210 (table 3): I think it's important to discuss the species that showed up in the structure survey that didn't in the biological survey

Lines 218-220: Not sure this format of questioning is customary

Line 240: Recommend making this table a figure

Line 264: Move reference to the end of the sentence

Line 277: Remove commas after 'they' and 'too'

Line 278: Change 'biased' to "bias"

Line 284: 'It that' - there is a missing or wrong word here

Line 310: This is the first time invertebrates are mentioned - I don't think it is appropriate to say this.

6. PLOS authors have the option to publish the peer review history of their article (what does this mean?). If published, this will include your full peer review and any attached files.

Reviewer #1: No

Reviewer #2: No

---

## [Author Response · Author response to Decision Letter 0]

9 Sep 2020

Response to Reviewer’s Commonts of: 

A Comparison of Two ROV Survey Methods Used to Estimate Fish Assemblages and Densities Around a California Oil Platform

Authors: Love, Milton S., Mary M. Nishimoto, Scott Clark, Li Kui, Aivy Aziz, David Palandro

Thanks to the reviewers’ comments and suggestions, which greatly improve the quality of the manuscript. Please see the responses below. 

Reviewer #1: Review of PONE-D-20-16316 A comparison of two ROV survey methods used to estimate fish assemblages and densities around a California Oil Platform

General:

The paper provides a succinct assessment of how dedicated biological surveys of fish communities on a platform jacket compare to data that could be derived from typical industry surveys. The results are valuable as they demonstrate the utility of industry surveys and how slight adjustments to protocol can yield important biological data. It is also rare for such surveys to be conducted, with an effective doubling of sampling effort required by industry to undertake this research. I note the paper is focussed on fish, but examination of epibiota may also yield interesting results.

The paper is generally well written and clear with just a few grammatical errors (see below). I have made some specific suggestions below to improve the manuscript. In particular, further explanation of the differences in methodologies is required and the resulting implications for the study.

Specific:

The addition of a Figure that shows the reader what the jacket structure looks like (surface-seabed with crossbeams) would be a good addition and could be incorporated into Figure 2.

Done

Table 1: Include crossbeam length into the text within the table to make this clear.

Sorry we don’t understand this request.

Line 88 – Add a sentence explaining why these depths were chosen. Is this the only depths that crossbeams were evident? There is a big gap between 61 and 182 and 182 and the seabed.

We have added the following: “As we did not have sufficient funding to include all of the crossbeams in this survey, we surveyed midwater crossbeams of representative depths along with the crossbeam at the bottom.”

Methods in general: I would like to see more information on the differences between the methods. Did the structural survey use lights? How did the speed compare between both surveys? 

We have inserted the following: “All surveys were conducted at the same speed using the SubC 1CamMk5 HDf video camera using lights at all depths..”

Also, would a structural survey undertake transects in the same manner as that used by the biological survey, i.e. of these same components or did they only survey these same crossbeams for comparison to the biological surveys? This is important because structural surveys may not survey the entire jacket but just sections of interest to industry.

Yes, the more complete surveys would cover the same crossbeams.

Line 104 - A 2 m distance for a biological survey (focused on fish) seems very close. McLean et al. (2019) used 5 m to provide an optimal view of the fish assemblage around the jacket. Do you feel that you missed larger species?

An interesting point. The fish community that McLean et al. (2019) surveyed is quite different from that found off California. McLean’s community is a tropical one, with substantial numbers of larger, much more motile jacks and snapper. Our community is composed, in midwaters, primarily of juvenile rockfishes, that form more-or-less slow to respond aggregations and, at the bottom, again of rockfishes and other relatively small, and importantly sedentary taxa. In addition, most of the fishes are either so small or so difficult to distinguish from congenerics that distances greater than about 2 m would mean that identifications would be either extremely problematic or impossible.

Line 138: I would stick with the raw numbers here rather than inflating by x100. Just adjust the scales on Figures so that the pattern is evident and present as per m2. This would mean altering Figures 4, 5 to present the correct values per m2. I really cant see this inflation as being necessary.

We understand your concerns, but we also prefer to keep the inflated density number in this manuscript. Here are some conventions: 1) Other researchers studying California platform fishes, i.e. Martin and Lowe (2009), have also used the same metric. 2) There is a convention in reporting densities of California reef and soft sea floor fishes to use the expanded area (x100). 3) all of our other platform studies are in those units, so we need this consistency in order for us, and for anyone else, to be able to compare the results in this study to those of other platform studies.

Line 145 – missing a word “..we created a two-dimensional..”

Done.

Line 147: spelling. Abundance should be abundant.

Done.

Table 2: Add “structural” to the industry column for consistency with text. Same for Table 3 and 

Table 4. I also note that the figures present structural first then biological. You could keep the tables in the same order.

Done.

Results – add that three species were uniquely observed by the biological survey and two by the structural survey, albeit each in very low numbers.

 We have the following: Two “species”, Sebastes flavidus/serranoides and Rhinogobiops nicholsii (along with a handful of unidentified fishes), were only observed using the biological method, and two species, Sebastes carnatus and Sebastes serriceps were unique to the structural surveys. All were observed in very small numbers.

Line 180 – should this be 26.9 not 27.1? (to match Table 2).

We changed to 26.9.

Line 183: So here you are stating that whichever survey method occurred first was more likely to view these aggregations of young rockfishes. Likely because they were scared away or avoided the ROV subsequently. The following sentence needs further justification then, i.e. for each of the three sides of the jacket or days did you start with a different survey method? (randomised order).

We have made substantial changes to the text and added a new Table 2 to help explain both the methods (order of surveys at each crossbeam) and the results. 

In the Introduction we have added the following: All surveys were conducted during daylight hours, and decisions regarding which side was surveyed first, which method was first used on a specific side, and the length of time between surveys on a specific side were made haphazardly (Table 2). 

 In addition, in the Results section we clarified the role that multiple passes may have on densities estimates: “Although mean densities derived from the biological method were higher than from the structural method (35.2 m-2 and 26.9 m-2, respectively), they were not statistically different (p >0.05) (Table 3, Fig 4). Particularly, high densities occurred at three crossbeams. When we more closely examined the video footage, we found that these extremely high densities were due to loose and moving aggregations of young rockfishes. These aggregations were present during one of the two survey techniques along the crossbeams (Points A, B, and C in Fig 4), and then densities along the same crossbeams were substantially lower on the other after the aggregations moved away. Importantly, Point A represents a second pass and Points B and C, first passes. Therefore, to the extent that we can determine, in these instances differences in densities were not due to the difference in ROV survey method, but rather to other, unknown, environmental parameters.”

Figure 4 legend: Make it clear in the legend that densities have been inflated (x100)

We add a sentence in the caption to clarify that. “Fish density values were multiplied by 100 to obtain density units of fish per 100 m2.”

Line 202 – make sure numbers match the table. They are slightly off.

Done. 

Line 204 – do you mean that the highest density estimate was obtained on the north side using the biological method? This is unclear here? If referring to Point D, what is the actual species that is driving this? Is it Sebastes melanostomus?

We inserted the following text to clarify the issue: “This highest fish density was observed on the north side of the platform jacket using the biological method, and was driven primarily by two species groups: unidentified Sebastes with a density of 23 m-2 and Sebastes melanostomus, with a density of 13 m-2. By comparison, the lowest density estimate was observed using the industrial method on the north side, and the two survey techniques yielded similar density estimates on the south and west sides of the platform jacket (Fig 5).”

Line 205 – change industrial to “structural”

Done. 

Table 4 legend is missing a close bracket.

 The errant parenthesis was removed.

Line 284: missing word. “It showed that..”

Done. We used “found” rather than “showed.”

Line 293: Great! Bring on HOV surveys into the future.

Ha! If only we could…

Line 295: California’s platform jackets – plurals needed.

Done.

Line 310: remove invertebrates from the sentence.

Done.

Line 312: This sentence does not make sense or is missing a few words.

Done.

Reviewer #2: Love and colleagues compared fish assemblage and density data derived from two types of ROV surveys along an ExxonMobil Platform off the coast of southern California. The goal was to determine if video observations from routine approaches used in the industry could be utilized by scientists to monitor fish populations, as is being done by others elsewhere (e.g., Gulf SERPENT project). The first approach was one used by the industry to examine structural integrity (structure) and involved variable camera angles along cross-beams. The second, used more by scientists (biological), involved a 90 degree scan and incorporated the seabed on the bottom cross-beam. Overall, the authors identified community similarities between the two techniques, suggesting a potential benefit of the industry-based methods, although there were significant differences when considering communities at the base of the platform.

The main weakness of the manuscript is that it is based on so few comparative surveys (n=12) between the two approaches, which are preformed at a single platform, spread across different depths of the water column and different "faces" of the platform jacket. It is understood that these surveys are quite costly to implement, but the lack of statistical rigor should be further emphasized. Further, there is a tremendous amount of detail lacking that could affect the interpret-ability of the results. As the authors themselves admit, fish can become deterred by ROV presence. So, a major question becomes: how were these surveys performed (in terms of timing) and could one have impacted the results of the other (e.g., was biological always before industry)? Was randomization possible? All we are provided with are the dates over which the surveys were completed, and that they were done by the same ROV type. It was also clear that the authors were not well-versed in their community analyses as there are terms used incorrectly, and again were lacking details on how they set up their statistical analysis. I have identified these shortcomings in the attached PDF.

We certainly recognize that the sample size was relatively small, and the survey time was short, which were due to the expense of this kind of survey. We have added several sentences in the discussion to emphasize the limitation of this study. For instance, we begin the penultimate paragraph “Due to the small sample size (n=12) for each of the survey methods and short survey time (2 days), only the simple statistics were computed in this study to explore the overall difference between these two survey methods.” The relevant statistical method descriptions have also been revised to clarify the analysis. 

Stylistically, the manuscript is well-written and the figures strongly convey the results in a readable manner. However, there were several typos throughout, which have also been highlighted in my review.

In all, I believe Love et al. have some intriguing preliminary evidence that industry-type ROV surveys could provide meaningful data to be used by scientists for monitoring fish densities and assemblages at oil platforms. I am not sure that the findings are particularly "earth-shattering" given the limited number of surveys in a constrained time-period, and feel this manuscript would probably be better suited as a rapid communication.

Specific Comments (by line numbers):

ABSTRACT

Line 13: Abstract - this is way too long. It shouldn't exceed 300 words

Done. Now 294 words.

INTRODUCTION

Line 67: Consider adding reference:

Ajemian, M. J., Wetz, J. J., Shipley-Lozano, B., & Stunz, G. W. (2015). Rapid assessment of fish communities on submerged oil and gas platform reefs using remotely operated vehicles. Fisheries Research, 167, 143-155.

Done.

Line 79: Strange return

MATERIALS AND METHODS

Line 87: Which survey was first?

See the responses given for Reviewer 1 re line 183.

Line 112: List camera type and specs

We included this in a sentence around lines 129–131.

Line 131: Need more details on the chronology of the surveys

See the responses given for Reviewer 1 re line 183.

Line 131: More details on whether all footage was analyzed or if there was a sub-sampling regime

We added “All footage, along all crossbeams, was included in the analyses.”

Line 141: Wouldn't this be just a mixed-model ANOVA?

Yes. We changed this in the statistical analysis section. 

Line 144-147: Need to state data type used in analysis (density, presence/absence)

It is the density value and we added it into the manuscript. The sentence now reads “The response variable was a matrix of fish density of the top ten most abundant species that consisted of 83% of the total fish count.”

Line 147: Replace 'abundance' with "abundant"

Thanks! We corrected the spelling. 

Line 150: Change 'Dissimilarity' to "similarity”

Done 

Line 151: What type of matrix was used?

Species density matrix. We added it into the manuscript. The sentence now reads “To statistically test whether the species assemblages between survey methods and habitats were different, we used the Analysis of Dissimilarity (ANOSIM), the anosim() function in “vegan” package, following by calculating the matrix of species dissimilarity using vegdist() function.”

RESULTS AND DISCUSSION:

Lines 166-168: The NMDS is simply an ordination; it does not give you a significance level. Is this from the ANOSIM?

Yes, it is from the ANOSIM. We added in more detail: “However, the ANOSIM analysis for the survey method was not significant (p = 0.69), suggesting that survey method was not likely to contribute to the species assemblage difference.”

Line 169: Change 'MNDS' to "NMDS"

Done.

Lines 183-186: This is exactly why the chronology of the surveys needs explanation in the methods.

See the responses given to Reviewer 1, line 183.

Line 197: For 'primarily,' did you mean "presumably" - it does not make sense otherwise

Done.

Line 210 (table 3): I think it's important to discuss the species that showed up in the structure survey that didn't in the biological survey

The fishes that were not held in common were observed in very small, and likely no meaningful, numbers. We have added the following to the results: “Two “species”, a single Eptatretus stouti and an unidentified Nettastomatidae, were unique to the biological survey and three “species”, an unidentified flatfish, a Sebastomus sp., and three Leuroglossus stilbius individuals, were only see in the structural survey. These were all observed in very small numbers.”

Lines 218-220: Not sure this format of questioning is customary

We would prefer to keep as is, as we consider that the wording assists the reader.

Line 240: Recommend making this table a figure

While the numbers in this table are presented in Figure 5, we believe the reader is best served through this table.

Line 264: Move reference to the end of the sentence

Done.

Line 277: Remove commas after 'they' and 'too'

Done.

Line 278: Change 'biased' to "bias"

Done.

Line 284: 'It that' - there is a missing or wrong word here

We added “found” to make “It found that”

Line 310: This is the first time invertebrates are mentioned - I don't think it is appropriate to say this.

 Removed.

---

## [Decision Letter · Decision Letter 1]

22 Sep 2020

PONE-D-20-16316R1

A comparison of two remotely operated vehicle (ROV) survey methods used to estimate fish assemblages and densities around a California oil platform

PLOS ONE

Dear Dr. Love,

Thank you for submitting your manuscript to PLOS ONE. After careful consideration, we feel that it has merit but does not fully meet PLOS ONE’s publication criteria as it currently stands. Therefore, we invite you to submit a revised version of the manuscript that addresses the points raised during the review process.

Academic Editor

This paper will be acceptable for publication when you have addressed the two minor points raised by the reviewer.

We look forward to receiving your revised manuscript.

Kind regards,

Maura (Gee) Geraldine Chapman, PhD DSc

Academic Editor

PLOS ONE

Additional Editor Comments (if provided):

Academic Editor

This paper will be acceptable for publication when you have addressed the two minor points raised by the reviewer.

Reviewers' comments:

Reviewer's Responses to Questions

**Comments to the Author**

1. If the authors have adequately addressed your comments raised in a previous round of review and you feel that this manuscript is now acceptable for publication, you may indicate that here to bypass the “Comments to the Author” section, enter your conflict of interest statement in the “Confidential to Editor” section, and submit your "Accept" recommendation.

Reviewer #2: All comments have been addressed

2. Is the manuscript technically sound, and do the data support the conclusions?

Reviewer #2: Yes

3. Has the statistical analysis been performed appropriately and rigorously? 

Reviewer #2: Yes

4. Have the authors made all data underlying the findings in their manuscript fully available?

Reviewer #2: Yes

5. Is the manuscript presented in an intelligible fashion and written in standard English?

Reviewer #2: Yes

6. Review Comments to the Author

Reviewer #2: Thank you for making the various revisions to the manuscript. It seems that the majority of the recommended changes have been incorporated and the manuscript is vastly improved.

A few final (minor) comments:

ABSTRACT: Thanks for reducing this, although there is a return in there (not sure if that was accidental) so it seems like two paragraphs still.

Line 178: Remove comma after 'particularly'

7. PLOS authors have the option to publish the peer review history of their article (what does this mean?). If published, this will include your full peer review and any attached files.

Reviewer #2: No

---

## [Author Response · Author response to Decision Letter 1]

13 Oct 2020

This is the final version of:

A Comparison of Two Remotely Operated Vehicle (ROV) Survey Methods Used to Estimate Fish Assemblages and Densities Around a California Oil Platform

Love, M. S., M. M. Nishimoto, S. Clark, L. Kui, A. Aziz, D. Palandro

In response to the comments of reviewer #2:

1) I have changed the abstract, so that it is only a single paragraph.

2) I have removed the comma after “particularly” in line 177.

Milton Love

Research Biologist

Marine Science Institute

University of California

Santa Barbara, California

---

## [Editor Report · Decision Letter 2]

26 Oct 2020

A comparison of two remotely operated vehicle (ROV) survey methods used to estimate fish assemblages and densities around a California oil platform

PONE-D-20-16316R2

Dear Dr. Love,

We’re pleased to inform you that your manuscript has been judged scientifically suitable for publication and will be formally accepted for publication once it meets all outstanding technical requirements.

Kind regards,

Maura (Gee) Geraldine Chapman, PhD DSc

Academic Editor

PLOS ONE
---

## [Editor Report · Acceptance letter]

29 Oct 2020

PONE-D-20-16316R2 

A comparison of two remotely operated vehicle (ROV) survey methods used to estimate fish assemblages and densities around a California oil platform 

Dear Dr. Love:

I'm pleased to inform you that your manuscript has been deemed suitable for publication in PLOS ONE. Congratulations! Your manuscript is now with our production department. 

Kind regards, 

on behalf of

Professor Maura (Gee) Geraldine Chapman 

Academic Editor

PLOS ONE